# Low-Temperature Processed Brookite Interfacial Modification for Perovskite Solar Cells with Improved Performance

**DOI:** 10.3390/nano12203653

**Published:** 2022-10-18

**Authors:** Jiandong Yang, Jun Wang, Wenshu Yang, Ying Zhu, Shuang Feng, Pengyu Su, Wuyou Fu

**Affiliations:** 1State Key Laboratory of Superhard Materials, Jilin University, Changchun 130012, China; 2School of Physics and Optoelectronic Engineering, Shandong University of Technology, Zibo 255000, China; 3College of Mathmatics and Physics, Inner Mongolia Minzu University, Tongliao 028000, China; 4School of Electronic Information Engineering, Yangtze Normal University, Chongqing 408100, China

**Keywords:** brookite, nanorods, pore filling, perovskite solar cell, interfacial modification, energy band

## Abstract

The scaffold layer plays an important role in transporting electrons and preventing carrier recombination in mesoporous perovskite solar cells (PSCs), so the engineering of the interface between the scaffold layer and the light absorption layer has attracted widespread concern. In this work, vertically grown TiO_2_ nanorods (NRs) as scaffold layers are fabricated and further treated with TiCl_4_ aqueous solution. It can be found that a thin brookite TiO_2_ nanoparticle (NP) layer is formed by the chemical bath deposition (CBD) method on the surface of every rutile NR with a low annealing temperature (150 °C), which is beneficial for the infiltration and growth of perovskite. The PSC based on the TiO_2_ NR/brookite NP structure shows the best power conversion of 15.2%, which is 56.37% higher than that of the PSC based on bare NRs (9.72%). This complex structure presents an improved pore filling fraction and better carrier transport capability with less trap-assisted carrier recombination. In addition, low-annealing-temperature-formed brookite NPs possess a more suitable edge potential for electrons to transport from the perovskite layer to the electron collection layer when compared with high-annealing-temperature-formed anatase NPs. The brookite phase TiO_2_ fabricated at a low temperature presents great potential for flexible PSCs.

## 1. Introduction

Thanks to the advantages of high absorption coefficients, broad absorption range, tunable bandgaps and long diffusion lengths, as well as the low-cost solution-processable method, the perovskite solar cell (PSC) is regarded as one of the most promising candidates for third-generation photovoltaic technologies [1,2,3,4,5,6,7,8]. After intensive efforts in interface engineering, solvent engineering, composition engineering, etc., the power conversion efficiency (PCE) of PSCs has increased from an initial 3.8% in 2009 to 25.2%, which makes it possible for PSCs to be applied in commercialization in the future [9,10,11,12,13,14,15,16]. A typical perovskite solar cell is composed of an electron transport layer (ETL), light absorption layer, hole transport layer (HTL) and counter-electrode. In addition, with the difference in the structure of the ETL, PSCs can be divided into planar and mesoporous PSCs. Compared with the planar structure, the mesoporous structure is most commonly used in PSCs owing to its slight hysteresis effect, effective carrier extraction capability as well as high stability [17,18,19]. A mesoporous-structure ETL contains a compact blocking layer and a scaffold layer. More importantly, the scaffold layer has been proven crucial for high-performance PSCs [20,21]. Thus, any changes or replacements in this layer will significantly affect the final performance of devices.

Inorganic metal oxides, such as aluminum oxide (Al_2_O_3_) [22], zinc oxide (ZnO) [23], zirconium oxide (ZrO_2_) [24], tin oxide (SnO_2_) [25] and titanium dioxide (TiO_2_) [26], as the scaffold layer materials, are most commonly applied to extract photogenerated electrons from the perovskite light absorption layer to the compact layer. Among all of the materials mentioned above, TiO_2_, with the advantages of simple fabrication, stable crystal structure as well as high transparency, is widely employed in the state-of-the-art PSCs. Miyasaka fabricated mesoporous TiO_2_ thin films for CH_3_NH_3_PbBr_3_ and CH_3_NH_3_PbI_3_ solar cells and obtained a PCE of 3.13% and 3.81%, respectively [10]; Park’s group achieved a PCE of 9.7% by decreasing the thickness of mesoporous TiO_2_ and replacing the liquid electrolyte with a solid electrolyte (spiro-OMeTAD) [27]. For TiO_2_ nanorod (NR)-based PSCs, Park investigated different lengths of rutile TiO_2_ NRs, and an enhanced PCE from 5.9% to 9.4% was obtained [28]; Wongratanaphisan investigated the influence of the annealing temperature on the TiO_2_ NR scaffold layer and obtained a 15.5% PCE with the TiO_2_ NR layer treated with boiling water [29]. However, there is a non-negligible problem that PSCs based on bare NRs often suffer from internal pore filling, which will result in low light absorbance, poor interfacial transfer and unwanted carrier recombination [30,31]. To solve this problem, Haining deposed a tiny anatase TiO_2_ nanoparticle (NP) layer to modify the surfaces of NRs by the liquid phase deposition (LPD) method and obtained a PCE of 8.61% [32]; Sawanta S prepared an ultrathin atomic layer to passivate the TiO_2_ NRs’ surface and obtained an enhanced PCE of 13.45% [33]. Besides the methods mentioned above, TiCl_4_ treatment is an efficient way to optimize the interface between ETL and perovskite. An nm-thick layer of TiO_2_ NP is formed at the surface of NRs by the CBD method in a TiCl_4_ aqueous solution, which improves the interface and creates efficient charge transfer access from the perovskite layer to ETL. However, most reports use this method with high-temperature annealing to obtain an anatase NP layer [31,34].

In this work, brookite NPs are successfully synthesized using the CBD method with a low annealing temperature on the surface of a rutile NR. Such engineering leads to a good pore filling fraction and uniform perovskite film and obtains the best PCE of 15.2%. In addition, the influence of annealing temperature on the formation of TiO_2_ was investigated. Compared with the anatase phase formed at a high temperature, low-temperature-fabricated brookite-phase TiO_2_ exhibits better carrier transport capability.

## 2. Experiment

### 2.1. Materials

PbI_2_ (99.99%), CH_3_NH_3_I (≥99.5%), and spiro-OMeTAD (≥99.5%) were purchased from Xi’an Polymer Light Technology Corp. (Xi’an China) with no purification. TiCl_4_ (99%), N,N-Dimethylformamide (DMF, chromatographic grade, ≥99.9%), methanol (chromatographic grade, *≥*99.9%), and isopropanol (≥99.5%) were purchased from Aladdin (Shanghai, China). Hydrochloric acid (36.5–38.0 wt%) was purchased from Xilong Chemical Co. Ltd. (Guangdong, China). Tetrabutyl titanate was purchased from Tianjin Berens Biotechnology Co. Ltd. (Tianjin, China). The fluorine-doped tin oxide coated glass (FTO, 6 ohm sq^−1^) was purchased from Opvtech New Energy Co., Ltd. (Yingkou, China).

### 2.2. Device Fabrication

The following procedures were all performed in a fume hood without any manipulation of the ambient temperature, humidity or airflow. F-doped SnO_2_ glasses were etched with Zn powder and aqueous HCl solution, and then ultrasonically cleaned with soap, deionized water, acetone, isopropanol and ethanol in sequence, followed by a 15 min UV–O_3_ treatment. The compact layer was fabricated through the chemical bath deposition (CBD) method in 100 mM aqueous solution of TiCl_4_ at 70 °C for 30 min. TiO_2_ NRs (approximately 400 nm) were grown on the as-fabricated substrates by a hydrothermal method, similar to a previous report [35]. Briefly, 30 mL distilled water was poured slowly into 30 mL hydrochloric acid (36.5–38.0 wt%) solution. Then, 1 mL tetrabutyl titanate was dropped into the mixture with constant stirring for 20 min. The mixed solution and the as-prepared TiO_2_ layer were placed into a sealed teflon reactor (100 mL volume) for hydrothermal growth at 170 °C for 100 min. The samples were washed with deionized water and dried in the air when cooled down to room temperature. Then, samples were annealed at 450 °C for 30 min. For TiCl_4_-treated NRs, the as-prepared samples were placed into a 0.02 M TiCl_4_ aqueous solution at 70 °C for 30 min. Then, the samples were annealed at 150 °C, 300 °C and 450 °C for 30 min, respectively, to obtain TiO_2_ NPs with different crystalline phases. We also treated the NRs in 0.4 M TiCl_4_ aqueous solution at 70 °C for 60 min with 150 °C annealing to measure the phase of the NPs.

Next, 1.2 M PbI_2_ solution (552 mg PbI_2_ in 1 mL N,N-dimethylformamide (DMF) was firstly dropped upon the electron transport layer (ETL) at 3000 rpm for 30 s, and then CH_3_NH_3_I solution (30 mg/mL in isopropanol) was spin-coated onto the PbI_2_ layer at 5000 rpm for 30 s. The resultant perovskite film was subsequently annealed at 150 °C for 15 min. The hole-transporter layer was prepared by spin-coating spiro-OMeTAD solution at 3000 rpm for 10 s onto the perovskite layer. Finally, an 80 nm Ag layer via thermal evaporation was deposited onto the hole-transporter layer as a counter-electrode.

### 2.3. Characterization

The phase of synthesized samples was obtained by an X-ray diffractometer (Rigaku D/max-2500) with a 1.5418 Å Cu Kα line. The morphology and microstructure of samples were assessed by scanning electron microscopy (SEM, FEI Magellan 400). Atomic force microscopy (AFM) images were measured by an XE-7 scanning probe microscope (Park Systems, Suwon, Korea). To further demonstrate the structure of brookite nanoparticles, transmission electron microscopy (TEM) images and high-resolution TEM (HRTEM) images were obtained by a JEM-2100F microscope. Ultraviolet–visible (UV–vis) absorbance spectra were measured via a spectrophotometer (UV-3150 double-beam). Steady-state photoluminescence (PL) diagrams were acquired by an HR Evolution Raman spectrometer (excitation of 473 nm at room temperature). Photocurrent density–photovoltage curves were measured by a Keithley 2400 Source Meter, and the simulated sunlight illumination (AM 1.5 G, 100 mW/cm^2^) was provided by a 500 W xenon lamp system (CLE-S500) with a BG26M92C laser power meter to calibrate. The active area of fabricated PSCs was controlled by a mask at around 0.1 cm^2^. Serial resistance and recombination resistance were obtained from electrochemical impedance spectroscopy (EIS) spectra using an electrochemical workstation (ZAHNER IM6, Kronach, Germany) whose frequency range was from 0.1 Hz to 100 kHz.

## 3. Results and Discussion

It is well known to all that some pores existing in the interior or interface between the ETL and perovskite layer could result in the poor photovoltaic performance of PSCs. Therefore, a sufficient pore filling fraction is crucial for a good PSC. Figure 1a depicts the XRD patterns of the TiO_2_ NR/perovskite film and TiO_2_ NR/brookite NP/perovskite film to study the effect of the brookite NPs on the crystallization of the perovskite film. Both of the samples showed three main perovskite diffraction peaks at 14.02°, 28.4° and 32.0°, which correspond to the (110), (220) and (310) crystal faces. Compared with the TiO_2_ NRs/perovskite film, the TiO_2_ NR/brookite NP/perovskite film presents stronger diffraction intensity, and the full width at half maximum (FWHM) diffraction peak of CH_3_NH_3_PbI_3_ is smaller than that of the TiO_2_ NR/perovskite sample, which reveals that perovskite can obtain better crystallinity after NRs are treated with TiCl_4_. Meanwhile, an obvious diffraction peak at around 12.7°, which resulted from the residual PbI_2_ in the perovskite film, disappeared in the TiO_2_ NR/brookite NP/perovskite sample. This phenomenon indicates that TiCl_4_ treatment can achieve a better reaction from PbI_2_ and CH_3_NH_3_I to CH_3_NH_3_PbI_3_. Moreover, to further identify the phases of the TiO_2_ NRs and NPs, we measured the XRD patterns of bare NRs, NRs treated with 0.02 M TiCl_4_ and NRs treated with 0.4 M TiCl_4_, which are presented in Figure 1b. As can be seen from this figure, the rutile phase, which belongs to TiO_2_ NRs, can be observed both in bare NRs and TiCl_4_-treated samples at the diffraction peak at 36.08°. Expectably, there is no diffraction peak of NPs either in the bare NR sample or NRs treated with 0.02 M TiCl_4_, which is mainly owing to the low quantity of NPs below the XRD detection limits. However, when the concentration of TiCl_4_ is increased to 0.4 M, there are new diffraction peaks existing at 25.69° and 40.15°, which belong to the brookite phase. The XRD result indicates that brookite TiO_2_ NPs can be obtained by the CBD method with low-temperature annealing.

Figure 2a,b,d,e show the cross-sectional SEM of the TiO_2_ NR film, TiO_2_ NR/perovskite film, TiO_2_ NR/brookite NPs film and TiO_2_ NR/brookite NP/perovskite film. The top-view SEM images of the TiO_2_ NR/perovskite film and TiO_2_ NR/brookite NP/perovskite film are presented in Figure 2c,f. It is obvious that TiO_2_ NRs grow vertically on the fluorine-doped tin oxide (FTO). After spin-coating perovskite, some pores, as carrier recombination centers, which will have a detrimental effect on the final performance of solar cells, can be observed (Figure 2b). Moreover, there are some obvious “white phase” PbI_2_ nanocrystals existing in the perovskite film (Figure 2c) [36,37]. As can be seen from Figure 2d, brookite NPs form a conformal layer at the surface of the NR after treatment by TiCl_4_. The conformal layer could form a barrier layer at the interface between perovskite and FTO to suppress charge recombination [34]. After sequential two-step spin-coating of perovskite, we notice that perovskite materials infiltrate into NRs’ interspaces, with almost no pores or PbI_2_ remnants (Figure 2e,f). The SEM images prove that the growth and formation of perovskite can be influenced by the morphology of the scaffold layer.

Atomic force microscopy (AFM) was used to investigate the surface roughness of the bare TiO_2_ NR film and TiO_2_ NR/brookite NP film, which are shown in Figure 3. The root-mean-square (RMS) surface roughness of the bare TiO_2_ NRs film is 21.2 nm, which is smaller than that of the TiO_2_ NR/brookite NP film (23.8 nm). The increased roughness could be attributed to the brookite NPs, which will form a rough surface for PbI_2_ to load around the NRs and further form the perovskite layer.

TEM and HRTEM were used to characterize the morphology and crystal structure of the TiO_2_ NR and TiO_2_ NR/brookite NPs. As shown in Figure 4a,b, the bare NR possesses a smooth surface with a length of around 400 nm. Moreover, the lattice fringes of bare NRs with an interplanar spacing d = 0.324 nm, which corresponds to the (110) plane of rutile TiO_2_, can be observed in Figure 4b. This pure rutile phase indicates that the NR is a single-crystal structure with high crystallinity, which could provide direct access for photo-generated electrons to transport from the perovskite layer to the electron collecting layer. Apart from the bare rutile NR, the surface of the NR in Figure 4c is covered by NPs with a thickness of 5 nm. These NPs increase the roughness of the NR and offer a nanostructured interfacial contact between perovskite and ETL, which will promote carrier separation and transport. In addition, we notice that, besides the lattice distances of 0.324 nm, there are also existing lattice spacing values of 0.346 nm and 0.225 nm, which correspond to the (111) plane and (112) plane of brookite TiO_2_ (JCPDS No. 29-1360) (Figure 4d). These crystalline planes and lattice spaces demonstrate that TiCl_4_ treatment with low-temperature annealing produces a tiny uniform brookite NP layer, which covers the surface of the rutile TiO_2_ NR.

UV–vis absorption was used to assess the absorption properties of perovskite samples with or without brookite NPs. The resulted absorption spectra (Figure 5a) reveal that both samples exhibit the same absorption onset (approximately 800 nm), yet the perovskite based on the TiO_2_ NR/brookite NP film shows increased absorption strength, which further verifies the better growth of the perovskite film as analyzed in SEM. Moreover, the charge transport processes of the perovskite deposited on TiO_2_ NRs and TiO_2_ NR/brookite NP films were examined by steady-state PL spectra, as shown in Figure 5b. The FTO/TiO_2_ NR/brookite NP/perovskite sample shows lower quenching than that of the FTO/TiO_2_ NR/perovskite sample. This means that the TiO_2_ NR/brookite NP structure has better capability for electron extraction and suppresses non-radiative decay [38], which is in accordance with the SEM images and photoelectric parameters. Figure 5c depicts the optimal photocurrent–voltage curves of PSCs based on TiO_2_ NRs and TiO_2_ NR/brookite NP ETLs under one sun AM 1.5G irradiance, and their photoelectric parameters are summarized in the figure. The power conversion efficiency (PCE) of bare NRs is 9.72%, with a short-circuit current density (J_sc_) of 19.18 mA/cm^2^, an open-circuit voltage (V_oc_) of 0.962 V and a fill factor (FF) of 0.52, while the TiO_2_ NR/brookite NP-based device shows enhanced photoelectric performance with a PCE of 15.2%, J_sc_ of 21.34 mA/cm^2^, V_oc_ of 1.05 V and FF of 0.67. The enhanced V_oc_ and FF could be ascribed to the less trap-assisted recombination. Meanwhile, the increased J_sc_ could be attributed to the better infiltration and growth of perovskite materials. To further reveal the carrier transport as well as recombination process, EIS measurements are carried out with a bias of 0.8 V. The Nyquist plots of two samples are shown in Figure 5d. The equivalent circuit is inserted in the figure to fit the data, where R_s_ is the serial resistance and R_rec_ is the recombination resistance of electrons in the interface between perovskite and ETL. Both R_s_ and R_rec_ can be obtained by fitting the impedance data with an equivalent circuit, and the larger semi-circle reflects higher recombination resistance and more efficient charge transport [39]. According to Figure 5d, the R_rec_ of the TiO_2_ NR/brookite NP sample (8203 Ω) is larger than that of the bare TiO_2_ NR sample (1336 Ω), which indicates that brookite NPs have a positive influence on suppressing trap-assisted carrier recombination.

Whereas other works treated the TiO_2_ NPs with a high annealed temperature to obtain anatase-phase NPs, we investigated the TiO_2_ NRs with NPs annealed at different temperatures to further study the influence of temperature on NPs. Figure 6a shows the UV–vis absorption spectrum of bare TiO_2_ NRs and TiCl_4_-treated NRs with annealing temperatures of 150 °C, 300 °C and 450 °C, respectively. The optical absorption ranges of the four samples are in the ultraviolet region. Although TiCl_4_ treatment has little influence on the optical absorption range of the TiO_2_, the absorption strength of TiCl_4_-treated NRs is higher than that of bare NRs. This might be ascribed to the NPs with different annealing temperatures affecting the phase formation of TiO_2_ and then further changing the band gap. The corresponding band gap (Eg) of bare rutile TiO_2_ NRs and TiCl_4_-treated NRs with annealing temperatures of 150 °C, 300 °C and 450 °C can be calculated by the Kubelka–Munk equation [40,41], which is presented in Figure 6b. The Eg of untreated TiO_2_ NRs and TiCl_4_-treated NRs with annealing temperatures of 150 °C, 300 °C and 450 °C can be determined to be 3.22 eV, 3.17 eV, 3.20 eV and 3.19 eV. TiCl_4_-treated NRs with an annealing temperature at 150 °C show the narrowest energy band gap, which is similar to other reports [42].

Figure 7a–d show the statistical values for the photovoltaic parameters of 15 independent PSCs, which are based on the four different ETLs, and Table 1 provides their relevant average photovoltaic parameters. According to the outcomes, the average FF, J_sc_ and V_oc_ of TiCl_4_-treated NR samples with different annealing temperatures are higher than those of bare TiO_2_ NRs, which is mainly owing to the NRs after TiCl_4_ treatment possessing a rough surface, which is beneficial for perovskite materials to infiltrate into the interspaces of NRs and decrease carrier recombination centers [43]. However, with the annealing temperature exceeding 150 °C, the performance of cells decreases. It might be attributed to the gradual transition of TiO_2_ NPs from brookite to anatase, since brookite TiO_2_ has a more suitable energy band structure than anatase, allowing for the more efficient transfer of photo-generated electrons from the perovskite layer to the rutile NRs.

The band alignment and charge transport behavior in the TiO_2_ NR/TiO_2_ NP-based PSCs are illustrated in Figure 1. As can be seen, electron–hole pairs are separated in the perovskite layer under illumination, and then electrons can transfer from the perovskite layer to the TiO_2_ layer while holes transfer to HTL. For the poor performance of PSCs fabricated on bare NRs, the rutile-phase NRs with a single-crystal structure could provide direct accesses for photo-generated electrons to transfer. However, there are some pores that are caused by the smooth surface of the NR, existing at the interface of the perovskite layer and the NR layer. Meanwhile, photo-generated electrons are difficult to transfer to NRs due to the most negative conduction band (CB) minimum of rutile TiO_2_ when compared to anatase and brookite [44,45,46]. The above reasons will finally result in poor performance. On the contrary, for PSCs fabricated on TiCl_4_-treated NRs, the existence of NPs increases the surface coverage and thus facilitates the interfacial contact of perovskite/TiO_2_. In the electron transfer process, rutile NR/brookite NP and rutile NR/anatase structures provide an energy level cascade, which could promote electron transfer. Since brookite NPs possess a more negative CB minimum with respect to that of anatase NPs, such an energetic structure provides better access for photo-generated electrons to transfer from perovskite to the CB of brookite NPs and then to the CB of rutile NRs. Therefore, the performance of brookite NP-based PSCs compared with anatase NP-based PSCs can be enhanced.

## 4. Conclusions

In summary, a tiny brookite NP interfacial modification layer is successfully fabricated by the CBD method at a low annealing temperature. SEM images and PL spectra confirm that perovskite could achieve better growth and the problem of pores causing recombination is obviously improved; meanwhile, the rutile NR/brookite NP structure could form a suitable energy level cascade for electrons to transfer. PSCs based on the as-fabricated NR/brookite NP structure obtained the best PCE of 15.2% with J_sc_ = 21.34 mA/cm^2^, V_oc_ = 1.05 V and FF = 0.67, which are higher than those of bare NR-based devices. Such low-temperature-fabricated brookite nanocrystals are promising for band alignment and the cost-effective preparation of perovskite solar cells.

## Data Availability

Not applicable.

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
