# Peer review of "Low-Temperature Processed Brookite Interfacial Modification for Perovskite Solar Cells with Improved Performance"

_nanomaterials, 2022, doi:10.3390/nano12203653_

Round 1
Reviewer 1 Report
In page one authors say "the best power conversion of 15.3% which is 57% higher than that of bare NRs based PSCs". Give references here to be able to compare the efficiences with bare NRs.
Please tell clearly what is the siez of the fabricated solar cells, as without this the presented results are meaningless. A photo of a fabricated solar cell would also be welcome.
The conclusions conclusions concerning the photoluminescence results are also mistaken - we can only compare the samples which photoluminescent intensity was measured using an integrating sphere, which is not the case here. The differences in the thicknesses (due to the differences in the structure of each sample) are also to be taken into account.
What is the largest surface that can be covered using the CBD technique to fabrucate brookite TiO2 NPs (for future industrial applications)?
Reviewer 2 Report
1. In 2.2, indicate what is a light source and what is a meter since it is not clear from the phrase: "the simulated sunlight illumination was carried out using a BG26M92C laser power meter".
2. In Figure 5a, indicate the optical density not in arbitrary units in order to make clear how the absorption changes in the processed TiO2.
3. Since the brookite NP layer noticeably changes the absorption (Fig. 5 a, b), it is recommended to speculate about the thickness of the NP layer.
4. On the plot presented in Fig.6b, it is necessary to justify the use of (αhν) to the power of 2. Since TiO2 is known as an indirect band gap semiconductor so power ½ should be applied [J. Phys. Chem. Lett. 2018, 9, 23, 6814–6817].
5. Check the grammar of the text. On p.5, for example, there is no predicate in the following sentence: "Besides, the lattice fringes of bare NR with an interplanar spacing d = 0.324 nm, corresponding to the (110) plane of rutile TiO2."
Reviewer 3 Report
This work shows TiO2 nanorods with a post treatment creating what the authors state is brookite phase of TiO2 as nanoparticles on the exterior of rutile nanorods. The treatment which creates the nanoparticles aids in the quality of the perovskite material. The authors show this improvement of the perovskite as a reduction of the PbI2 peak found in XRD and increase in the absorbance and PL of the perovskite.
The paper shows convincing evidence that the perovskite is improved. But what is lacking is the proof of the phases and characterization of the TiO2 and the energy alignment to explain why the device performance is improved.
The JV curves of the device show irregularities (most notably the dip seen in both devices at 0.5 V in the JV scan). This indicates some barrier to extraction present in both devices, or the effects of hysteresis from scanning too quickly, etc. However, the device performance isn't the major component of the paper.
The identification of the brookite vs rutile and anatase is lacking. Additionally the energy band diagram (even if the energy levels are believed) doesn't explain specifically why one device would be better than the other. The band edge positions could use better convincing data to justify.
Round 2
Reviewer 2 Report
Comments to responses:
1. OK
2. In Fig. 5a, there is no perovskite at all. Instead of a.u. represent the Y axis in units of the measured optical density A. The molar absorption coefficient is not needed.
3. OK
4. OK
5. OK
Reviewer 3 Report
The revisions look appropriate
